# Synthesis, Structure, Spectral-Luminescent Properties, and Biological Activity of Chlorine-Substituted *N*-[2-(Phenyliminomethyl)phenyl]-4-methylbenzenesulfamide and Their Zinc(II) Complexes

**DOI:** 10.3390/ijms232315259

**Published:** 2022-12-03

**Authors:** Anatolii S. Burlov, Valery G. Vlasenko, Maxim S. Milutka, Yurii V. Koshchienko, Nadezhda I. Makarova, Vladimir A. Lazarenko, Alexander L. Trigub, Alexandra A. Kolodina, Alexander A. Zubenko, Anatoly V. Metelitsa, Dmitrii A. Garnovskii, Alexey N. Gusev, Wolfgang Linert

**Affiliations:** 1Institute of Physical and Organic Chemistry, Southern Federal University, 344090 Rostov-on-Don, Russia; 2Institute of Physics, Southern Federal University, 344090 Rostov-on-Don, Russia; 3National Research Centre “Kurchatov Institute”, 123182 Moscow, Russia; 4North-Caucasian Zonal Scientific Research Veterinary Institute, Branch of the Federal State Budget Scientific Institution “Federal Rostov Agricultural Research Centre”, 346421 Novocherkassk, Russia; 5Southern Scientific Center of Russian Academy of Sciences, 344006 Rostov-on-Don, Russia; 6General Chemistry Department, Crimean Federal University V.I. Vernadsky, 295007 Simferopol, Russia; 7Institute of Applied Physics, Vienna University of Technology, Wiedner Hauptstraße 8-10, 1040 Vienna, Austria

**Keywords:** azomethines, zinc(II) complexes, photoluminescence, electroluminescence, biological activity

## Abstract

New azomethine compounds of 2-(*N*-tosylamino)benzaldehyde or 5-chloro-2-(*N*-tosylamino)benzaldehyde and the corresponding chlorine-substituted anilines, zinc(II) complexes based on them have been synthesized. The structures of azomethines and their complexes were determined by elemental analysis, IR, ^1^H NMR, X-ray spectroscopy, and X-ray diffraction. It is found that all ZnL_2_ complexes have a tetrahedral structure according to XAFS and X-ray diffraction data. The photoluminescent properties of azomethines and zinc complexes in methylene chloride solution and in solid form have been studied. It is shown that the photoluminescence quantum yields of solid samples of the complexes are an order of magnitude higher compared to the solutions and range from 11.34% to 48.3%. The thermal properties of Zn(II) complexes were determined by thermal gravimetric analysis (TGA) and differential scanning calorimetry. The TGA curves of all the compounds suggest their high thermal stability up to temperatures higher than 290 °C. The electrochemical properties of all complexes were investigated by the cyclic voltammetry method. The multilayered devices ITO/PEDOT:PSS/NPD/Zn complex/ TPBI/LiF/Al with wide electroluminescence (EL) color range spanning the range from bluish-green (494 nm) to green (533 nm) and the high values of brightness, current and power efficiency were fabricated. The biological activity of azomethines and zinc complexes has been studied. In the case of complexes, the protistocidal activity of the zinc complex with azomethine of 5-chloro-2-(*N*-tosylamino)benzaldehyde with 4-chloroaniline was two times higher than the activity of the reference drug toltrazuril.

## 1. Introduction

For a long time, the interest of researchers in the synthesis and study of the physicochemical properties of new Schiff bases by salicylaldehyde derivatives and transition metal complexes based on them has not faded [1,2,3]. Significant progress in this area is primarily due to the practical importance of these compounds. In particular, these compounds demonstrate a wide range of biological activity [4,5] exhibiting antifungal [6,7], antibacterial [6,7,8], anticancer [9,10,11,12,13], and antiviral [14] properties, which determines their demand by medicine, veterinary medicine, and agriculture.

Particular attention is drawn to zinc complexes of chelating azomethine ligands, owing to their photo- and electroluminescent properties, which allow their use as emitting layers in OLED devices, which have many advantages in terms of power consumption, brightness, and color compared to traditional sources [15,16,17,18,19,20,21]. These complexes have a number of advantages, having high thermal stability, volatility, and solubility; they are relatively simple for synthesis, non-toxic and cheap. In addition, varying the nature of substituents in azomethine ligands makes it possible to fine-tune the luminescent characteristics of zinc complexes based on them. For example, based on Zn(II) complexes containing salicylaldiminate ligands, blue, greenish-white, and red emitters for organic photoelectronics with good stability and efficiency have been obtained [22,23]. Of particular interest are the Zn(II) complexes of azomethine ligands of 2-(*N*-tosylamino)benzaldehyde [24,25] and 3-methyl-1-phenyl-4-formylpyrazol-5-one [26,27,28,29,30] derivatives. Replacing the hydroxy group in the aldehyde fragment of azomethines with a 2-tosylamino group with the formation of the MN_4_ coordination site leads to an increase in thermal stability (m.p. increases by about 50–100 °C) [7,24,31,32,33] and solvent resistance of complexes as compared to azomethine complexes based on salicylic aldehydes. In addition, the introduction of chlorine atoms into position 5 of the 2-(*N*-tosylamino)benzaldehyde fragment promotes an increase in the solubility of both azomethines and complexes based on them. Zinc complexes based on azomethine ligands of 2-(*N*-tosylamino)benzaldehyde fluoresce in the blue region of the spectrum λ_PL_ = 428–433 nm and have a high fluorescence quantum yield φ > 0.20, which makes them promising objects as an active emission layer of OLED.

Previously [34,35,36], we obtained a series of zinc complexes of bidentate azomethine ligands of halogenated 2-hydroxybenzaldehydes derivatives, and studied their structure, spectral-luminescent properties, and biological activity. In continuation of [35,36], we synthesized a similar series of azomethine compounds of 2-(*N*-tosylamino)benzaldehyde or 5-chloro-2-(*N*-tosylamino)benzaldehyde and the corresponding chlorine-substituted anilines, and zinc(II) complexes based on them. The main goal of this work was to study the effect of the halogen substituent in the amine and aldehyde fragments in zinc complexes on their photoluminescent properties and biological activity.

## 2. Results and Discussion

### 2.1. Synthesis of Azomethines and Their Zinc Complexes

Azomethine compounds **1a**–**e** were obtained by condensation of 2-(*N*-tosylamino)benzaldehyde or 5-chloro-2-(*N*-tosylamino)benzaldehyde and the corresponding chlorosubstituted anilines in glacial acetic acid (Figure 1).

### 2.2. Elemental Analysis and Spectroscopic Data

The structure of azomethines **1a**–**e** was established by elemental analysis, IR, and ^1^H NMR spectroscopy. All of them are finely crystalline substances of white color with m.p. from 133 °C to 182 °C. The IR spectra of compounds **1a**–**e** show absorption bands ν(CH=N) in the region 1614–1620 cm^−1^, ν(asSO_2_) in the region 1317–1343 cm^−1^ and ν(sSO_2_) at 1155–1162 cm^−1^.

The ^1^H NMR spectra of azomethines **1a**–**e** contain signals of all protons corresponding to their structures. The signals of the protons of the NH groups of the aldehyde fragments appear as a singlet in the range of 11.62–12.57 ppm, and the signals of the protons of the CH=N groups in the range of 8.62–8.76 ppm.

The synthesis of zinc bis-chelates **2a**–**e** was carried out by refluxing the corresponding azomethine **1a**–**e** and a methanol solution of zinc acetate dihydrate in a 2:1 molar ratio in a mixture of methanol: chloroform (1:1) (Figure 1). All complexes are yellow crystalline substances with high melting points from 273 °C to 311 °C. According to the elemental analysis data, zinc complexes **2a**–**e** have the composition ZnL_2_. In the IR spectra of zinc complexes **2a**–**e**, the absorption bands ν(CH=N) are observed at 1605–1611 cm^−1^, shifting to the low-frequency region by 3–13 cm^−1^, compared to the **1a**–**e**, and the absorption bands ν(asSO_2_) and ν(sSO_2_) are shifted to the low-frequency region by 22–43 cm^–1^ to 1285–1302 cm^–1^ in the case of ν(asSO_2_) and by 19–23 cm^–1^ to 1131–1143 cm^–1^ in the case of the band ν(sSO_2_). In the ^1^H NMR spectra of complexes **2a**–**e**, the proton signals of the NH groups of **1a**–**e** disappear. The proton signals of CH=N groups in **2a**–**e** are slightly shifted downfield by 0.02–0.15 ppm compared to **1a**–**e** and appear at 8.74–8.78 ppm. Such changes observed in the IR and ^1^H NMR spectra of zinc complexes **2a**–**e** are characteristic of the formation of chelate structures [24,25,31,34].

### 2.3. X-ray Absorption Spectroscopy

The local atomic structure of the nearest atomic environment of zinc ions in complexes **2a**–**d** was established from X-ray absorption spectroscopy data from the XANES and EXAFS analysis of Zn K absorption edges. Figure 1 shows normalized XANES and the corresponding MFT EXAFS for all compounds. It can be noted that the positions and shapes of the Zn K absorption edges **2a**–**d** are very close to each other (Figure 1a), indicating a similar environment of zinc ions in these compounds. The XANES of complexes **2a**–**d** lack the pre-edge peak A due to the filled 3d shell of Zn(II). The intense peak C (white line) corresponds to the maximum of the X-ray absorption spectrum and has the same energy position for all K absorption edges of complexes **2a**–**d**.

Quantitative characteristics of the coordination polyhedron in **2a**–**d** were obtained from the EXAFS analysis of the Zn *K*-edges. Figure 1b shows the modules of Fourier transform (MFT) of EXAFS of these compounds. All MFTs have the main peak at r = 1.51–1.53 Å, which corresponds to the scattering of a photoelectron wave by the nearest coordination sphere (CS) of the nitrogen atoms of the ligands. The MFT peaks at large values of r > 2.5 Å are associated with subsequent CSs containing various ligand atoms, mainly the carbon atoms, as well as oxygen and sulfur of the tosylamine fragments of the ligands. As a result of the performed EXAFS model calculations, it was found that the nearest environment of zinc ions in all complexes **2a**–**d** is very close and consists of four nitrogen atoms with average distances Zn…N about 1.92 Å, Zn…N about 2.01 Å (Table 1). The obtained values of the Debye-Waller factors of about 0.0030 Å^2^ are typical for such Zn…N distances in coordination compounds with similar structures [36].

### 2.4. A Single-Crystal X-ray Diffraction

The structures of **2d** and **2a** complexes were established by X-ray diffraction. The structure of complexes **2a**, **2d,** and the structure of crystal **2a** along the [100] axis are shown in Figure 2. The complexes **2a** and **2d** with the metal-to-ligand ratio 1:2 are described by an idealized C_2_ (2) symmetry and crystallize in the C_2_/c space group with the chemical formulae C_40_H_32_Cl_2_N_4_O_4_S_2_Zn and C_40_H_28_Cl_6_N_4_O_4_S_2_Zn, respectively. The Zn ions have an oxidation state of 2+ and bicapped tetrahedral coordination environment “4 + 2” by four N atoms from the tosilamino and imine groups [Zn1-N1 1.9763(14) Å, Zn1-N2 2.0507(14) Å for **2a** and Zn1-N1 1.98763(3) Å, Zn1-N2 2.04076(3) Å for **2d**], and additional weaker interactions with two O atoms from sulfo groups [Zn1···O1 2.66919(13) Å for **2a** and Zn1···O1 2.67283(4) Å for **2d**]. The average values of bond lengths coincide with the average values of such bonds from the CSD [37]. The crystal packings of analyzed complexes are enhanced by intermolecular hydrogen bonds O1···H6′-C6′/O1′···H6-C6 and O2···H9′-C9′/O2′···H9-C9 for **2d/2a**, respectively. The distances and angles for these hydrogen bonds for **2d/2a** are [C···O 3.17985(4)/3.199(2)Å, H···O 2.40149(4)/2.3041(13)Å, ∠C-H···O 138.9652(8)/161.19(11)°] for the first pair of hydrogen bonds, and [C···O 2.99820(6)/3.078(2)Å, H···O 2.35724(4)/2.4076(15)Å, ∠C-H···O 124.3966(10)/128.91(11)°] for the second pair. The crystal structure of **2a** (Figure 3) is strengthened by π-π interactions between benzene rings of the tosyl fragments of ligands with a 3.927 Å centroid-centroid distance and 1.904 Å shift distance.

### 2.5. The Electron Absorption and Photoluminescence Spectra

The absorption and photoluminescence (PL) spectra of zinc complexes **2a**–**e** in methylene chloride solution (Figure 4) and in solid form (Figure 5) were studied. In the UV-Vis spectra of complexes **2a**–**e** in a solution of CH_2_Cl_2_, two absorption bands are observed in the region λ_max_ = 306–312 nm and λ_max_ = 385–400 nm (Table 2). When excited into long-wavelength absorption bands, fluorescence is observed with band maxima in the region λ_PL_ = 488–509 nm (Figure 4). The Stokes shift is between 103 and 197 nm. The quantum yields φ of PL in CH_2_Cl_2_ solutions are insignificant and range from φ = 0.029 to φ = 0.062. As can be seen from Table 2, the introduction of a second chlorine atom into the aniline fragment of the ligand leads to a slight increase in the quantum yield of PL. The maxima of the PL bands of solid samples of complexes are shifted to the long-wavelength region of the spectrum compared to solutions. The PL quantum yields of solid samples of the complexes are an order of magnitude higher than those of solutions and range from 11.34% to 48.3%. The introduction of a chlorine atom into position 5 of the aldehyde fragment leads to the bathochromic shift of the fluorescence band of solid samples of the complexes and a decrease in the fluorescence quantum yield (Table 2).

It should be noted that compared with zinc complexes based on 5-chloro-2-hydroxybenzaldehyde [35], the replacement of the hydroxy group of the aldehyde fragment with the tosylamino group (**2c**) leads to a hypsochromic shift of the PL bands by 4 nm (in CH_2_Cl_2_ solution) and 23 nm (solid-state) and an increase in PL quantum yields from 2% to 5% (in CH_2_Cl_2_ solution) and from 14.7% to 21.4% (solid-state).

### 2.6. Thermal and Electrochemical Properties

The thermal stability of luminescent complex is very important for OLED applications, which are directly related to devise performance such as efficiency, lifetime, etc. The thermal properties of title complexes **2a**–**e** were determined by thermal gravimetric analysis (TGA) and differential scanning calorimetry (DSC) under a nitrogen atmosphere. The main parameters of thermal transitions are shown in Table 3. The TGA curves of all compounds suggest high thermal stability estimated by the start of the decomposition of complexes (Td, corresponding to 2% weight loss) (Appendix A). No loss of adsorbed or coordinated solvate molecules was observed for all complexes, which is consistent with EA data. The complexes only had a 2% weight loss at 340 (**2a**), 305 (**2b**), 335 (**2c**), and 290 (**2d**) °C. Complexes **2b** and **2d** having substitutions in the *meta*-position of aldehyde moiety, exhibited lower Td. Upon further heating, decomposition starts above 390–420 °C with a series of complicated weight losses, which continue until heating ends at 670–700 °C. Detailed decomposition mechanisms of the four complexes are too complex to be explained. The remaining weight agrees with the theoretical value calculated by taking zinc oxide as the final product.

The first DSC heating scans confirmed that complexes were obtained as crystalline materials with melting point (Tm) values of 317, 304, 311, and 275 °C for **2a**–**d,** respectively. The melting of the title complexes exhibits sharp endothermic peaks. No transition peaks associated with crystallization peaks were observed during the DSC cooling scans, which indicated the formation of glasses. In the second DSC heating scan, only complex **2a** exhibited glass transitions with glass transition temperatures (Tg) 101 °C. There was no evident glass-transition temperature (Tg) observed from 50 °C to 150 °C for the other compounds, which can be attributed to the more rigid structure due to strong intermolecular interactions. The excellent thermal properties of **2a**–**d** inspired us to fabricate efficient OLEDs by vacuum thermal evaporation technique.

Further, we used cyclic voltammetry to investigate the electrochemical properties and determine the energy levels of title complexes. The electrochemical behavior was studied in dry acetonitrile (0.01 M solutions) by cyclic voltammetry (CV) measurement in a three-electrode cell system. The detailed electrochemical properties of the Zn(II) complexes are listed in Table 3. During the anodic scan, a monoelectronic and irreversible anodic wave with a peak maximum at 1.04–1.11 V corresponding to the oxidation of the Schiff base moieties is observed in the CV profiles. From the electrochemical and spectral data, the energies of the LUMO and HOMO levels of **2a–d** were estimated as listed in Table 3. According to the oxidation onset-potential values, the highest unoccupied molecular orbital (HOMO) energy levels E_HOMO_ were estimated to be −5.21–5.33 eV. The relative lower E_HOMO_ of **2a** and **2c** originated from the stronger electron-withdrawing ability of chlorine-atom in *meta*-position. Estimation of the LUMO energy values was carried out by the addition of an energy gap calculated from spectral data. LUMO levels are located in the range of −2.59 to −2.79 eV.

### 2.7. Electroluminescence Studies

In order to evaluate the electroluminescent (EL) properties of the **2a**–**d** complexes, the following multilayered devices ITO/PEDOT:PSS/NPD/**2a**–**d**/ TPBI/LiF/Al. were fabricated in which **2a-d** act as the emitters named: device A—**2a**; device B—**2b**; device C—**2c**; device D—**2d**, respectively. PEDOT:PSS (poly(3,4-ethylenedioxythiophene) polystyrene sulfonate)—hole injector. NPD (N,N′-Di(1-naphthyl)-N,N′-diphenyl-(1,1′-biphenyl)-4,4′-diamine)—hole transporter. TPBI 1,3,5-tris(N-phenylbenzimidazole-2-yl)benzene—electron transporter. LiF is used both to lower the energy barrier between the LUMO level of the Zn complex and the electron emission work from aluminum and to prevent Al aluminum atoms from entering the emitting layer when forming the cathode. Aluminum served as the cathode

All devices exhibited the typical diode characteristics. The light turn-on voltage for all devices was in the range of 4.5–7 V. All devices emitted a greenish-blue or green light (Figure 6a). Figure 6b shows the normalized EL spectra of title devices. In accordance with their PL spectra, a wide EL color range spanning from bluish-green (494 nm for **2a**) to green (533 nm for **2d**) was obtained for these devices. The EL spectra of these devices are almost invariant of the applied voltages, and it is clear that the EL spectra of the **2a–b** complex matched well with their PL in terms of spectral profile and position, confirming that no emission signal from electroplex or excimer/exciplex has appeared in these devices under electroexcitation. From the EL spectra, it was concluded that the peak position of the emitted light is independent of the applied voltage. Complexes **2a** and **2c,** which contain only one Cl-atom in the aldehyde fragment of the ligand, show better performance than complexes **2b** and **2d** which have two Cl substituents. For example, the luminescence intensity of device A-D increases with increasing voltage reaching a brightness of more than 1545 Cd/m^2^ at 18 V in the case of device A and 1007 Cd/m^2^ at 17 V in the case of device C, while the brightness of the other devices was below 1000 Cd/m^2^ at 18 V. Further increasing of bias voltage leads to fast destruction of cells. Figure 6c,d show the current-voltage and brightness-voltage characteristics of the created OLED structures, and Table 4 presents the electroluminescence parameters. Device A also shows the highest in this series maximum current and power efficiency of 2.7 cd/A, 1.7 lm/W. Maximum EQE was found to be 2.11 (A), 1,20 (B); 1,25 (C); 0,9 (D) %, respectively.

Although the luminance and efficiency values of the resulting devices do not reach the maximum values for close structures [29,30,31], these preliminary results suggest that the title complexes have the potential for use as emissive layers in organic light-emitting diodes. However, the device architecture needs to be improved, by using perhaps a different host matrix, with better electron-transporting properties. 

### 2.8. The Biological Activity

The biological activity of azomethine compounds **1a–e** and zinc(II) complexes **2a–e** based on them was studied. It was found that all azomethines **1a–e** and the zinc complexes **2a**–**e** show no fungistatic activity against *Penicillium italicum* (Table 5). Azomethines **1b**, **1d**, and **1e** show antibacterial activity against *Staphylococcus aureus*, but their activity is 2.2 times weaker than that of the reference drug furazolidone in the case of azomethines **1b**,**d** containing two chlorine atoms in the amine moiety, and 1.3 times weaker in the case of azomethine **1e** containing no chlorine atoms.

Only azomethines **1a** and **1e** are active against *Escherichia coli*. The activity of azomethine **1a**, containing a chlorine atom in the amine part, is 2.3 times, and the activity of **1e** is 1.2 times weaker than that of furazolidone. In the case of complexes, the complexes **2a**, **2c**, **2d**, and **2e** show bacteriostatic activity against *Staphylococcus aureus*, but the activity of complexes **2a**, **2c**, **2d**, containing from one to three chlorine atoms in the ligand, is 2.2–2.5 times and the activity of complex **2e**, which does not contain chlorine atoms in the ligand, is 1.3 times weaker than that of furazolidone. With respect to *Escherichia coli*, complexes **2c** and **2d** show activity 1.5–1.8 times, and complex **2e** is 1.3 times weaker than that furazolidone.

It was found that compound **1e**, containing no chlorine atoms, has high protistocidal activity among azomethines (Table 5). Its activity is 4 times higher than the activity of the reference drug toltrazuril, whereas azomethines **1a**–**d** containing chlorine atoms at different positions of azomethine, show weak protistocidal activity, which is 4–8 times weaker than the activity of the reference drug toltrazuril. Among the complexes **2a**–**e**, the most active against *Colpodasteinii* is the complex **2c**, containing one chlorine atom in each of the aldehyde and amine parts of the ligand, whose activity is 2 times stronger than that of toltrazuril. The activity of complex **2e** is 8 times weaker than that of toltrazuril. Complexes **2a**,**b**,**d** did not show protistocidal activity.

When comparing the antibacterial activity of azomethines **1a**–**e** and their complexes **2a**–**e**, it was shown that complexes **2a**,**c**, containing one or two chlorine atoms in the ligand, showed activity against *Staphylococcus aureus* 2.2–2.5 times weaker than the reference drug, in contrast to the corresponding azomethines, which did not have such activity at all. Complex **2d**, containing two chlorine atoms in the amine part of the ligand, showed no activity compared to azomethine **1d**, which has antibacterial activity against *Staphylococcus aureus*. The activities of complexes **2d**,**e**, and the corresponding azomethines **1d**,**e** was found to be the same. With respect to *Escherichia coli*, complexes **2c**,**d** containing two or three chlorine atoms in the ligand showed activity 1.5–1.8 times weaker than the reference drug, in contrast to the corresponding azomethines, which did not possess such activity at all. Complex **2a** showed no activity compared to azomethine **1a**, which has antibacterial activity against *Escherichia coli*. The activities of complex **2e** and the corresponding azomethine **1e** were found to be the same. Complex **2b** and azomethine **1b** did not have the corresponding activity.

When comparing the protistocidal activity of azomethines **1a**–**e** with their complexes **2a**–**e**, it was found that the protistocidal activity of the complexes decreases compared to azomethines, with the exception of complex **2c**, which contains one chlorine atom in each of the aldehyde and amine parts of the ligand. Complex **2c** showed activity 16 times greater than the activity of the corresponding azomethine **1c** and 2 times greater than the activity of the reference drug toltrazuril.

It should be noted that complexes **2a**,**b**, having one or two chlorine atoms in the amine part of the ligand, did not show protistocidal activity, whereas antibacterial activity among these complexes was observed only in monochlorine-substituted **2a** against *Staphylococcus aureus*, which was 2.5 times weaker than that of the reference drug. The introduction of a chlorine atom into the aldehyde part of the ligand of the complex with the retention of halogens in the amine part led to an increase in the biological activity of complexes **2c**,**d**, in particular, the zinc complex **2c** containing one chlorine atom in the aldehyde part and one chlorine atom in the amine part of the ligand, showed protistocidal activity 2 times higher than that of toltrazuril.

The replacement of a 2-hydroxybenzaldehyde fragment containing one or two chlorine atoms in azomethines with 2-(*N*-tosylamino)benzaldehyde or its 5-chloro derivative has no particular effect on the change in biological activity. This conclusion is consistent with previously published data for related azomethine compounds [38,39], where it was shown that the substitution of chlorine atoms in the aldehyde fragment of the 2-hydroxybenzaaldimine ligand of zinc complexes has a poor effect on their antimicrobial properties against Gram-positive (*Staphylococcus aureus*), and Gram-negative (*Escherichia coli*) organisms compared to the unsubstituted ligand, and less antimicrobial activity of known antibiotics (eg Gentamycin or Amoxicillin). The protistocidal activity of the zinc complex with azomethine of 5-chloro-2-(*N*-tosylamino)benzaldehyde with 4-chloroaniline was 2 times higher than the activity of toltrazuril, while the complex of zinc azomethine of 2-hydroxybenzaldehyde with 4-chloroaniline aniline did not exhibit this activity [35].

## 3. Materials and Methods

### 3.1. General Methods

Commercially available reagents were used as purchased without additional pretreatment and/or purification. The C, H, and N elemental analysis were carried out on a «EuroEA-3000» (EuroVektor) analyzer. The amount of the metal was determined by the gravimetric method. The melting temperature of the synthesized compounds was determined by the capillary method, and for complexes, **2a**–**2d** it was refined by DSC data. The IR spectra of the samples were recorded on a Varian 3100-FTIR Excalibur instrument in the range 4000–400 cm^–1^ by the method of disturbed total internal reflection. The ^1^H NMR spectra were recorded on a Varian Unity-300 instrument (300 MHz) in DMSO-*d*_6_. The chemical shifts of the ^1^H nuclei are given relative to the residual signals of the deuterated solvent. UV-Vis spectra were recorded for a 2.0 × 10^−5^ M solution with Agilent 8453 spectrophotometer. Photoluminescent spectra were recorded for a 5.0 × 10^−6^ M solution on a Varian Cary Eclipse fluorescence spectrophotometer. All spectra were recorded in dichloromethane (for spectroscopy, Acros Organics) solutions at room temperature. Fluorescence quantum yield determined relative to 3-methoxybenzanthrone in toluene as standard (φ = 0.1, excitation at 365 nm) [40]. Photoluminescent spectra in the solid state were recorded with Hamamatsu C11347-01 absolute PL quantum yield spectrometer. Absolute PL quantum yield was determined using an integrating sphere of the Hamamatsu C11347-01 spectrometer (excitation at 390 nm). TG and DSC experiments were performed on a STA6000 (PerkinElmer, Inc., Shelton, CT, USA) under a static nitrogen atmosphere. Cyclic voltammetry measurements for complexes **2a–d** were performed following the protocol described in ref [41].

The OLED devices were fabricated on glass substrates with 100 nm thickness ITO. Organic layers were sequentially deposited at a rate in the range of 0.1–0.3 nm/s onto the substrates by high-vacuum (10^−5^ mbar) thermal evaporation techniques. The shadow mask with 5 mm × 5 mm openings was used to define the cathodes. The evaporating speeds and thickness were monitored by quartz oscillators. All of the measurements were carried out in an ambient atmosphere at room temperature after a vacuum break. The current density–luminance–voltage characteristics of the OLEDs were measured by the Keithley source measurement unit with a calibrated silicon photodiode. Electroluminescence spectra were taken by a multichannel S2000 Ocean Optics spectrometer. All measurements were carried out in an ambient atmosphere at room temperature.

The X-ray Zn K absorption edges of zinc complexes were obtained in the transmission mode at the Structural Materials Science station at the Kurchatov Synchrotron Center (Moscow) [42]. The energy of the electron beam, which was used as a source of X-ray synchrotron radiation, was 2.5 GeV at an average current of 100–120 mA. The X-ray absorption spectra were processed by standard procedures for extracting the background, normalizing to the value of the K-edge jump, and extracting the atomic absorption μ_0_, after which the Fourier transform of the selected EXAFS (χ) spectrum was performed in the range of photoelectron wave vectors k from 2.5 to 12–13 Å^–1^ with the weight function k^3^. The exact values of the nearest environment parameters of the zinc ion in the studied compounds were determined by nonlinear fitting of the parameters of the corresponding coordination spheres when comparing the calculated EXAFS with that extracted by the Fourier filtering method from the full absorption spectrum. This procedure was carried out using the IFFEFIT software package [43]. The phases and amplitudes of photoelectron wave scattering required for constructing the model spectrum were calculated using the FEFF7 program [44]. As the initial atomic coordinates necessary for calculating the scattering phases and amplitudes and further fitting, we used X-ray diffraction data for single crystals of metal complexes with a similar molecular structure from the Cambridge database. The fit quality function Q, which was minimized when finding the structure parameters of the nearest environment, was calculated using the formula:$$Q^{2}=\frac{\sum_{i=1}^{m}w(k_{i})[k_{i}\chi_{\exp}(k_{i})-k_{i}\chi_{th}(k_{i}){]}^{2}}{\sum_{i=1}^{m}w(k_{i}){\left[k_{i}\chi_{\exp}(k_{i})\right]}^{2}}$$
where $w\left(k_{i}\right)$ is a weighting function, m is the number of experimental points, $\chi_{\mathrm{data}}\left(R_{i}\right)$ and $\chi_{\mathrm{th}}\left(R_{i}\right)$ are the EXAFS functions in the R-space.

The X-ray diffraction study of the crystals **2a** and **2d** was carried out at the “Belok/XSA” beamline of the Kurchatov Synchrotron Radiation Source [45,46]. Diffraction patterns were collected in direct geometry (θ = 0°) for **2a** and in a combination of direct and turned geometry (θ = 0° for the first data set and θ = 26° for the second data set) using a 1-axis MarDTB goniometer (φ-scanning mode with oscillation angle 1°) equipped with Rayonix SX165 CCD 2D positional sensitive CCD detector (λ = 0.745 Å). For the crystal of complex **2a,** data were obtained from two data sets, collected with different matrices of the orientation of the crystals, and for the crystal of complex **2d,** as described above. For each data set, ~180 diffraction frames were collected. All data were collected at 100 K.

The data were indexed and integrated by the XDS and XSCALE software suites [47]. The structures were solved by direct methods (intrinsic phasing) with SHELXT [48]. The structure model was investigated by Olex2 software [49] and refined by SHELXL [50] by a full-matrix least-squares method on F^2^ with anisotropic displacements. As a result of combining two sets of data obtained in different geometries for the crystal of the complex **2d**, the obtained data are characterized by a larger maximum angle of 2θ and, as a result, better resolution and greater accuracy of the structural model, which leads to lower anisotropic displacement parameters. The crystallographic parameters and the refinement statistics for **2a** and **2d** are given in Appendix A. Crystallographic data for **2a** and **2d** have been deposited with the Cambridge Crystallographic Data Center, CCDC 2203056 (**2a**), and CCDC 2203057 (**2d**). The supplementary crystallographic data can be obtained free of charge from the Cambridge Crystallographic Data Centre via www.ccdc.cam.ac.uk/data_request/cif (accessed on 7 October 2022.).

### 3.2. General Procedure for the Synthesis of Azomethines ***1a–e***

A hot solution of 10 mmol of chlorine-substituted aniline in 5 mL of glacial acetic acid was added to a hot solution of 10 mmol of 2-(*N*-tosylamino)benzaldehyde or 5-chloro-2-(*N*-tosylamino)benzaldehyde in 10 mL of glacial acetic acid, the reaction mixture was stirred for 1 h at 100 °C, then cooled to r. t. and 15 mL of ethanol were added. The precipitate that formed was filtered off, washed with ethanol, and dried in a vacuum oven at 100 °C.

*N-[2-[I-(4-Chlorophenyl)iminomethyl]phenyl]-4-methyl-benzenesulfonamide* (**1a**) was obtained from 2.75 g (10 mmol) of 2-(*N*-tosylamino)benzaldehyde and 1.28 g (10 mmol) of 4-chloroaniline. Yield 3.62 g (94%), white powder, m.p. 133–134 °C. IR spectrum, ν, cm^−1^: 1618 (CH=N), 1342 (asSO_2_), 1157 (sSO_2_). ^1^H NMR (DMSO-*d*_6_), δ, ppm: 2.29 s(3H, CH_3_), 7.18–7.23 m (1H, CH_arom_), 7.30–7.37 m (4H, CH_arom_), 7.41–7.46 m (2H, CH_arom_), 7.54 dd (2H, CH_arom_,3J = 6.9 Hz, 4J = 2.1 Hz), 7.67 d (2H, CH_arom_, 3J = 8.4 Hz), 7.76 d (1H, CH_arom_, 3J = 7.2 Hz), 8.74 s (1H, CH=N), 12.29 s (1H, NH). Anal. calc. for C_20_H_17_ClN_2_O_2_S, %: C 62.41; H 4.45; N 7.28. Found, %: C 62.44; H 4.48; N 7.31.

*N-[2-[(E)-(3,4-Dichlorophenyl)iminomethyl]phenyl]-4-methyl-benzenesulfonamide* (**1b**) was obtained from 2.75 g (10 mmol) of 2-(*N*-tosylamino)benzaldehyde and 1.62 g of (10 mmol) 3,4-dichloroaniline. Yield 3.90 g (93%), White powder, m.p. 139–140 °C. IR spectrum, ν, cm^−1^: 1616 (CH=N), 1343 (asSO_2_), 1160 (sSO_2_). ^1^H NMR (DMSO-*d*_6_), δ, ppm: 2.29 s (3H, CH_3_), 7.23–7.32 m (4H, CH_arom_), 7.39 d (1H, CH_arom_, 3J = 7.5 Hz), 7.45–7.48 m (1H, CH_arom_), 7.58 d (1H, CH_arom_, 4J = 2.4 Hz), 7.66 d (2H, CH_arom_, 3J = 8.4 Hz), 7.73 d (1H, CH_arom_, 3J = 8.7 Hz), 7.78 dd (1H, CH_arom_,3J = 7.8 Hz, 4J = 1.2 Hz), 8.71 s (1H, CH=N), 11.92 s (1H, NH). Anal. calc. for C_20_H_16_Cl_2_N_2_O_2_S, %: C 57.29; H 3.85; N 6.68. Found, %: C 57.25; H 3.81; N 6.64.

*N-[4-Chloro-2-[(E)-(4-chlorophenyl)iminomethyl]phenyl]-4-methyl-benzenesulfonamide* (**1c**) was obtained from 3.10 g (10 mmol) of 5-chloro-2-(*N*-tosylamino)benzaldehyde and 1.28 g (10 mmol) of 4-chloroaniline. Yield 3.77 g (90%), white powder, m.p. 181–182 °C. IR spectrum, ν, cm^−1^: 1617 (CH=N), 1337 (asSO_2_), 1162 (sSO_2_). ^1^H NMR (DMSO-*d*_6_), δ, ppm: 2.29 s (3H, CH_3_), 7.30–7.33 m (4H, CH_arom_), 7.39 d (1H, CH_arom_, 3J = 9.0 Hz), 7.52–7.56 m (3H, CH_arom_), 7.64 d (2H, CH_arom_, 3J = 8.1 Hz), 7.86 d (1H, CH_arom_, 4J = 2.4 Hz), 8.66 s (1H, CH=N), 11.96 s (1H, NH). Anal. calc. for C_20_H_16_Cl_2_N_2_O_2_S, %: C 57.29; H 3.85; N 6.68. Found, %: C 57.25; H 3.89; N 6.62.

*N-[4-Chloro-2-[(E)-(3,4-dichlorophenyl)iminomethyl]phenyl]-4-methyl-benzenesulfonamide* (**1d**) was obtained from 3.10 g (10 mmol) of 5-chloro-2-(*N*-tosylamino)benzaldehyde and 1.62 g (10 mmol) of 3,4-dichloroaniline. Yield 4.18 g (92%), white powder, m.p. 154–155 °C. IR spectrum, ν, cm^−1^: 1614 (CH=N), 1317 (asSO_2_), 1158 (sSO_2_). ^1^H NMR (DMSO-*d*_6_), δ, ppm: 2.28 s (3H, CH_3_), 7.25–7.32 m (3H, CH_arom_), 7.37 d (1H, CH_arom_, 3J = 9.0 Hz), 7.54 dd (2H, CH_arom_.,3J = 8.1 Hz, 4J = 2.4 Hz), 7.63 d (2H, CH_arom_, 3J = 8.4 Hz), 7.72 d (1H, CH_arom_, 3J = 8.4 Hz), 7.85 d (1H, CH_arom_, 4J = 2.7 Hz), 8.62 s (1H, CH=N), 11.62 s (1H, NH). Anal. calc. for C_20_H_15_Cl_3_N_2_O_2_S, %: C 52.94; H 3.33; N 6.17. Found, %: C 52.90; H 3.37; N 6.21.

*4-Methyl-N-[2-[(E)-phenyliminomethyl]phenyl]benzenesulfonamide* (**1e**) was prepared from 2.75 g (10 mmol) of 2-(*N*-tosylamino)benzaldehyde and 0.93 g (10 mmol) of aniline in 30 mL of ethanol. Yield 3.33 g (95%), Light yellow needles, m.p. 99–100 °C. IR spectrum, ν, cm^−1^: 2959 (NH), 1620 (CH=N), 1339 (asSO_2_), 1155 (sSO_2_). ^1^H NMR (DMSO-d_6_), δ, ppm: 2.29 s (3H, CH_3_), 7.17–7.23 m (1H, CH_arom_), 7.31–7.35 m (5H, CH_arom_), 7.45–7.51 m (4H, CH_arom_), 7.68 d (2H, CH_arom_, 3J = 8.1 Hz), 7.75 d (1H, CH_arom_, 3J = 7.5 Hz), 8.76 s (1H, CH=N), 12.57 s (1H, NH). Anal. calc. for C_20_H_18_N_2_O_2_S, %: C 68.55; H 5.18; N 7.99. Found, %: C 68.51; H 5.09; N 7.91.

### 3.3. General Procedure for the Synthesis of Complexes ***2a–e***

A solution of 0.22 g (1 mmol) of zinc acetate dihydrate in 5 mL of methanol was added to a boiling solution of 2 mmol of azomethine **1a–e** in 20 mL of a mixture of methanol and chloroform (1:1). The reaction was refluxed for 2 h, and then a solution of 0.08 g (2 mmol) of sodium hydroxide in 5 mL of methanol was added dropwise. The precipitate was filtered off, washed with methanol, and dried in a vacuum oven at 100 °C.

*Bis [2-[(E)-(4-chlorophenyl)iminomethyl]-N-(p-tolylsulfonyl)anilino]zinc(II)* (**2a**) was obtained from 0.77 g of **1a**. Yield 0.58 g (70%), yellow powder, m.p. 317 °C. IR spectrum, ν, cm^−1^: 1605 (CH=N), 1302 (asSO_2_), 1136 (sSO_2_). ^1^H NMR (DMSO-*d*_6_), δ, ppm: 2.30 s (3H, CH_3_), 6.92 t (1H, CH_arom_, 3J = 6.8 Hz), 7.16 d (2H, CH_arom_, 3J = 8.1 Hz), 7.33 d (4H, CH_arom_, 3J = 8.7 Hz), 7.42 d (2H, CH_arom_, 3J = 9.0 Hz), 7.60 d(2H, CH_arom_, 3J = 8.1 Hz), 7.74 d (1H, CH_arom_, 3J = 7.2 Hz), 8.76 s (1H, CH=N). Anal. calc. for C_40_H_32_Cl_2_N_4_O_4_S_2_Zn, %: C 57.67; H 3.87; N 6.72; Zn 7.85. Found, %: C 57.62; H 3.90; N 6.70; Zn 7.81.

*Bis[2-[(E)-(3,4-dichlorophenyl)iminomethyl]-N-(p-tolylsulfonyl)anilino]zinc(II)* (**2b**) was obtained from 0.84 g of **1b**. Yield 0.69 g (76%), yellow powder, m.p. 304 °C. IR spectrum, ν, cm^−1^: 1606 (CH=N), 1300 (asSO_2_), 1141 (sSO_2_). ^1^H NMR (DMSO-*d*_6_), δ, ppm: 2.30 s (3H, CH_3_), 6.96 t (1H, CH_arom_, 3J = 7.2 Hz), 7.19 d (2H, CH_arom_, 3J = 8.1 Hz), 7.25–7.39 m (3H, CH_arom_), 7.42 d (1H, CH_arom_, 4J = 1.8 Hz), 7.60 d (1H, CH_arom_, 3J = 8.4 Hz), 7.67–7.74 m (3H, CH_arom_,), 8.78 s (1H, CH=N). Anal. calc. for C_40_H_30_Cl_4_N_4_O_4_S_2_Zn, %: C 53.26; H 3.35; N 6.21; Zn 7.25. Found, %: C 53.21; H 3.39; N 6.23; Zn 7.29.

*Bis[4-chloro-2-[(E)-(4-chlorophenyl)iminomethyl]-N-(p-tolylsulfonyl)anilino]zinc(II)* (**2c**) was obtained from 0.84 g of **1c**. Yield 0.71 g (79%), yellow powder, m.p. 311 °C. IR spectrum, ν, cm^−1^: 1611 (CH=N), 1301 (asSO_2_), 1143 (sSO_2_). ^1^H NMR (DMSO-*d*_6_), δ, ppm: 2.30 s (3H, CH_3_), 7.17 d (2H, CH_arom_, 3J = 8.1 Hz), 7.29–7.33 m (3H, CH_arom_), 7.39 d (1H, CH_arom_, 4J = 2.4 Hz), 7.43 d (2H, CH_arom_, 3J = 8.7 Hz), 7.58 d (2H, CH_arom_, 3J = 7.8 Hz), 7.87 d (1H, CH_arom_, 4J = 2.7 Hz), 8.77 s (1H, CH=N). Anal. calc. for C_40_H_30_Cl_4_N_4_O_4_S_2_Zn, %: C 53.26; H 3.35; N 6.21; Zn 7.25. Found, %: C 53.22; H 3.39; N 6.25; Zn 7.22.

*Bis[4-chloro-2-[(E)-(3,4-dichlorophenyl)iminomethyl]-N-(p-tolylsulfonyl)anilino]zinc(II)* (**2d**) was obtained from 0.91 g of **1d**. Yield 0.71 g (73%), yellow powder, m.p. 275 °C. IR spectrum, ν, cm^−1^: 1608 (CH=N), 1295 (asSO_2_), 1135 (sSO_2_). ^1^H NMR (DMSO-*d*_6_), δ, ppm: 2.31 s (3H, CH_3_), 7.21 d (2H, CH_arom_, 3J = 8.1 Hz), 7.28 d (2H, CH_arom_, 3J = 9.3 Hz), 7.41 d (1H, CH_arom_, 4J = 2.4 Hz), 7.45 s(1H, CH_arom_), 7.60–7.67 m (3H, CH_arom_), 7.85 d (1H, CH_arom_, 4J = 2.4 Hz), 8.77 s (1H, CH=N). Anal. calc. for C_40_H_28_Cl_6_N_4_O_4_S_2_Zn, %: C 49.48; H 2.91; N 5.77; Zn 6.73. Found, %: C 49.43; H 2.95; N 5.75; Zn 6.78.

*Bis[2-[(E)-phenyliminomethyl]-N-(p-tolylsulfonyl)anilino]zinc(II)* (**2e**) was obtained from 0.70 g of **1e**. Yield 0.57 g (75%), light yellow powder, m.p. 295–296 °C. IR spectrum, ν, cm^−1^: 1609 (CH=N), 1285 (asSO_2_), 1131 (sSO_2_). ^1^H NMR (DMSO-*d*_6_), δ, ppm: 2.29 s (6H, CH_3_), 6.91–6.94 m (2H, CH_arom_), 7.14 d (4H, CH_arom_, 3J = 8.1 Hz), 7.31–7.33 m (14H, CH_arom_), 7.57 d (4H, CH_arom_, 3J = 7.5 Hz), 7.74 d (2H, CH_arom_, 3J = 7.8 Hz), 8.74 s (2H, CH=N). Anal. calc. for C_40_H_34_N_4_O_4_S_2_Zn, %: C 62.86; H 4.48; N 7.33; Zn 8.56. Found, %: C 62.69; H 4.52; N 7.40; Zn 8.61.

## 4. Conclusions

A number of azomethine compounds of 2-(*N*-tosylamino)benzaldehyde and 5-chloro-2-(*N*-tosylamino)benzaldehyde with chlorine-substituted anilines and their zinc(II) complexes were synthesized, the structure of which was determined by elemental analysis, IR, ^1^H NMR. According to the X-ray absorption spectroscopy data from EXAFS analysis of Zn K-edges, it was found that all complexes have a tetrahedral structure with similar values of Zn…O/N distances. The PL properties of azomethines and zinc complexes in methylene chloride solution and in solid form have been studied.

It is shown that Ihe introduction of a second chlorine atom into the aniline fragment of the ligand of zinc complexes leads to a slight bathochromic shift of both the UV-Vis absorption maximum and the PL maximum (by 0–3 nm). However, the introduction of a chlorine atom into the aldehyde fragment of the ligand of zinc complexes significantly shifts both the position of the absorption maximum of the UV-Vis spectrum and the maximum of PL by 12–15 nm to the red region. Quantum yields of PL of zinc complexes in the solid-state increase 4–12 times compared to the values in CH_2_Cl_2_ solutions.

Based on thermogravimetric analysis and differential scanning calorimetry, it was found that all complexes have high thermal stability (290–340 °C). The results of cyclic voltammetry made it possible to determine the energy levels of the complexes. The energy HOMO was −5.21–5.33 eV, with the lower values −5.21–5.33 eV (for **2a** and **2c**) arising from the stronger electron-withdrawing ability of the *meta*-position chlorine atom. TO/PEDOT:PSS/NPD/**2a**–**d**/ TPBI/LiF/AThe LUMO levels are in the range of −2.59 to −2.79 eV. The multilayered devices Il were fabricated in which **2a**–**d** acted as the emitters. The light turn-on voltage for all devices was in the range of 4.5–7 V. All devices emitted a greenish-blue (494 nm for **2a**) or green light (533 nm for **2d**). Complexes **2a** and **2c** with one Cl-atom in the aldehyde fragment of the ligand showed better performance than complexes **2b** and **2d** with two Cl substituents. The devices with **2a** emitter show the highest in this series maximum current and power efficiency of 2.7 cd/A, 1.7 lm/W.

The biological activity of azomethines and zinc complexes has been studied. Azomethins **1a**, **1b**, and **1d** had the highest antibacterial activity, but their activity was 2 times weaker than that of the reference drug furazolidone. In the case of complexes, the protistocidal activity of the complex of zinc with azomethine of 5-chloro-2-(*N*-tosylamino)benzaldehyde with 4-chloroaniline **2c** was two times higher than the activity of toltrazuril. The obtained results of the study of biological activity allow us to conclude that the search for antiprotozoal drugs among azomethines of halogen-substituted 2-(*N*-tosylamino)benzaldehyde with halogen-containing anilines and metal complexes based on them is promising.

## Data Availability

The data presented in this article are openly available.

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
