# Peer review of "Synthesis, Structure, Spectral-Luminescent Properties, and Biological Activity of Chlorine-Substituted N-[2-(Phenyliminomethyl)phenyl]-4-methylbenzenesulfamide and Their Zinc(II) Complexes"

_ijms, 2022, doi:10.3390/ijms232315259_

Round 1

Reviewer 1 Report (Previous Reviewer 2)

I am recommending  to accept the manuscript.

Author Response

IJMS-2041272, «Synthesis, structure, spectral-luminescent properties, and biological activity of chlorine-substituted N-[2-(phenyliminomethyl)phenyl]-4-methylbenzenesulfamide and their zinc(II) complexes»

Anatolii S. Burlov, Valery G. Vlasenko, Maxim S. Milutka, Yurii V. Koshchienko, Nadezhda I. Makarova, Vladimir A. Lazarenko, Alexander L. Trigub, Alexandra A. Kolodina, Alexander A. Zubenko, Anatoly V. Metelitsa, Dmitrii A. Garnovskii, Alexey N. Gusev and Wolfgang Linert

 ANSWERS TO REVIEWERS

We thank the Referees for their interest in our work and for helpful comments and constructive suggestions that will greatly improve the quality of this manuscript. As indicated below, we have checked all the general and specific comments provided by the Referees and have made necessary changes according to their indications (the reviewer’s comments are in italics).

Reviewers' comments:

Reviewer #1: I am recommending to accept the manuscript.

A: Many thanks to the reviewers for these positive feedbacks, which we appreciate.

Reviewer 2 Report (Previous Reviewer 3)

After reviewing the revision of this manuscript and reading the responses. I agree this manuscript can be accepted by IJMS.

Author Response

IJMS-2041272, «Synthesis, structure, spectral-luminescent properties, and biological activity of chlorine-substituted N-[2-(phenyliminomethyl)phenyl]-4-methylbenzenesulfamide and their zinc(II) complexes»

Anatolii S. Burlov, Valery G. Vlasenko, Maxim S. Milutka, Yurii V. Koshchienko, Nadezhda I. Makarova, Vladimir A. Lazarenko, Alexander L. Trigub, Alexandra A. Kolodina, Alexander A. Zubenko, Anatoly V. Metelitsa, Dmitrii A. Garnovskii, Alexey N. Gusev and Wolfgang Linert

ANSWERS TO REVIEWERS

We thank the Referees for their interest in our work and for helpful comments and constructive suggestions that will greatly improve the quality of this manuscript. As indicated below, we have checked all the general and specific comments provided by the Referees and have made necessary changes according to their indications (the reviewer’s comments are in italics).

Reviewers' comments:

Reviewer #2: After reviewing the revision of this manuscript and reading the responses. I agree this manuscript can be accepted by IJMS.

A: Many thanks to the reviewer for these positive feedbacks, which we appreciate.

Reviewer 3 Report (Previous Reviewer 1)

Please find comments to the manuscript ijms-2041272 (revised version of the manuscript ijms-1941620) 

I thank Authors for re-refinement of 2a-structure with a proper Z-value. Now, for both structures the empirical formulas are:

CCDC 2203057 (structure 2d) C40 H28 Cl6 N4 O4 S2 Zn  

CCDC 2203056 (structure 2a) C40 H32 Cl2 N4 O4 S2 Zn

My question: why are still  different (and strange) formulas in the manuscript at line 163? 

In lines 169-170 I propose to change the sentence as following:

"The crystal packings of analyzed complexes are enhanced by intermolecular hydrogen bonds ... "

Please omit 'four' interactions because four is a sum for both structures, each structure should be treated independently.

The molecular geometry is not supported by intermolecular interactions, the interactions are responsible for crystal structure.

Figure 3 was not corrected (the ellipsoids are still used instead of capped sticks or ball and stick mode). Authors proposed a new caption but the new one has more bugs. In Fig. 3 there is an arrangement of neighbouring molecules in the crystal of 2a showing the aromatic pi-pi interactions (it is not true that the molecules are within the unit cell, because some of them are clearly from other unit cells. Previously, I wrote that the true packing in the unit cell has involved more molecules. So the new caption will be true if  'in the unit cell' is removed. Moreover, in the first version of a caption was written  'along [100] axis' and it was in agreement with notation of crystallographic directions. Now, the view is proposed as 'in the  "perpendicular to the crystallographic direction (100)'. Direction in crystallography is, as it was stated before, in square brackets, perpendicular is not true, so please combined a caption to:

"Arrangement of neigbouring molecules in the crystal struture of 2a showing the aromatic pi-pi interactions; a view along the [100] direction".

Reference [50] should be removed. Citation [51] to SHELXL is enough and correct. 

Author Response

methylbenzenesulfamide and their zinc(II) complexes»

Anatolii S. Burlov, Valery G. Vlasenko, Maxim S. Milutka, Yurii V. Koshchienko, Nadezhda I. Makarova, Vladimir A. Lazarenko, Alexander L. Trigub, Alexandra A. Kolodina, Alexander A. Zubenko, Anatoly V. Metelitsa, Dmitrii A. Garnovskii, Alexey N. Gusev and Wolfgang Linert

ANSWERS TO REVIEWERS

We thank the Referees for their interest in our work and for helpful comments and constructive suggestions that will greatly improve the quality of this manuscript. As indicated below, we have checked all the general and specific comments provided by the Referees and have made necessary changes according to their indications (the reviewer’s comments are in italics).

Reviewers' comments:

Reviewer #3: I thank Authors for re-refinement of 2a-structure with a proper Z-value. Now, for both structures the empirical formulas are: CCDC 2203057 (structure 2d) C40H28Cl6N4O4S2Zn CCDC 2203056 (structure 2a) C40H32Cl2N4O4S2Zn. My question: why are still different (and strange) formulas in the manuscript at line 163?

A: We thank the reviewer #3 for these helpful comments.

In accordance with the referee' comment, we have now changed these empirical formulas in the manuscript at line 163.

Reviewer #3: In lines 169-170 I propose to change the sentence as following: "The crystal packings of analyzed complexes are enhanced by intermolecular hydrogen bonds ... " Please omit 'four' interactions because four is a sum for both structures, each structure should be treated independently. The molecular geometry is not supported by intermolecular interactions, the interactions are responsible for crystal structure.

A: Thank you for this suggestion. We have now changed this to "The crystal packings of analyzed complexes are enhanced by intermolecular hydrogen bonds ... " as suggested, which you will see on page 5, lines 169-170.

Reviewer #3: Figure 3 was not corrected (the ellipsoids are still used instead of capped sticks or ball and stick mode). Authors proposed a new caption but the new one has more bugs. In Fig. 3 there is an arrangement of neighboring molecules in the crystal of 2a showing the aromatic pi-pi interactions (it is not true that the molecules are within the unit cell, because some of them are clearly from other unit cells. Previously, I wrote that the true packing in the unit cell has involved more molecules. So the new caption will be true if  'in the unit cell' is removed. Moreover, in the first version of a caption was written  'along [100] axis' and it was in agreement with notation of crystallographic directions. Now, the view is proposed as 'in the  "perpendicular to the crystallographic direction (100)'. Direction in crystallography is, as it was stated before, in square brackets, perpendicular is not true, so please combined a caption to: "Arrangement of neighboring molecules in the crystal structure of 2a showing the aromatic pi-pi interactions; a view along the [100] direction".

A: The suggested correction of Figure 3 has been made. As suggested by the reviewer, we have revised a caption for Figure 3: "Arrangement of neighboring molecules in the crystal structure of 2a showing the aromatic π-π interactions; a view along the [100] direction".

Reviewer #3: Reference [50] should be removed. Citation [51] to SHELXL is enough and correct.

A: The suggested correction has been made.

This manuscript is a resubmission of an earlier submission. The following is a list of the peer review reports and author responses from that submission.

Round 1

Reviewer 1 Report

Comments to the manuscript IJMS-1941620:

- generally, a part describing the crystal structures 2a and 2d should be corrected and rewritten,

- the empirical formula (moiety formula)  (text in paragraph 2.4 and Table S1) is wrong for structures, it should be given not for asymmetric unit but for the whole complex-molecule (where Zn is 1),

- Z value in Table S1 (attention: wrong declared Z value in refinement of 2a) should be 4 instead of 8 because of a special position of Zn ion;

- structure 2a should be re-refined with Z=4 (as above),

- Rint and R (final indices) are mixed for structures; it should be checked and corrected,

- completeness for theta minimum is wrong for structure 2a, please check carefully (alert B in checkcif, it should be explained in cif submitted to deposit)

 - line 138 - the average values from CSD (not CCDC) should be given (and the citation to database)

- the hydrogen bonding parameters introduced in {} (lines 140-143) are difficult to read; it should be presented in more readable way,

- line 143-144 - please check and correct the description of pi-pi interaction in structure 2a,

- Figure 3 does not show the true packing along the crystallographic a axis (only 4 complex-molecules are used); the ellipsoids has no special meaning here, since they are defined in Fig. 2,

- the citation [44] for SHELXL is wrong.

Author Response

We thank the Referee for their interest in our work and for helpful comments and constructive suggestions that will greatly improve the quality of this manuscript. As indicated below, we have checked all the general and specific comments provided by the Referee and have made necessary changes according to their indications (the reviewer’s comments are in italics).

Reviewers' comments:

Reviewer #1: generally, a part describing the crystal structures 2a and 2d should be corrected and rewritten, the empirical formula (moiety formula)  (text in paragraph 2.4 and Table S1) is wrong for structures, it should be given not for asymmetric unit but for the whole complex-molecule (where Zn is 1)

A:The proposed changes have been made in text of paragraph 2.4 and Table S1

Reviewer #1: Z value in Table S1 (attention: wrong declared Z value in refinement of 2a) should be 4 instead of 8 because of a special position of Zn ion; structure 2a should be re-refined with Z=4 (as above)

A: The corrections have been made. The empirical formula is C40H32Cl2N4O4S2Zn with Z=4.

Reviewer#1: Rint and R (final indices) are mixed for structures; it should be checked and corrected

A: The corrections have been madein Table S1.Rint = 0.0596, R1 = 0.389, wR2 = 0.0989.

Reviewer #1: completeness for theta minimum is wrong for structure 2a, please check carefully (alert B in checkcif, it should be explained in cif submitted to deposit)

A: The new cif file for structure 2a submitted to CSD deposit. Low scattering power of the crystal, three data sets with different crystal orientation gave only 0.957 fraction. This is noted in the сif file.

Reviewer #1: line 138 - the average values from CSD (not CCDC) should be given (and the citation to database)

A: The average values from CSD are given in the text (C. R. Groom, I. J. Bruno, M. P. Lightfoot and S. C. Ward, ActaCryst. (2016). B72, 171-179).

Reviewer #1: the hydrogen bonding parameters introduced in {} (lines 140-143) are difficult to read; it should be presented in more readable way,

A: In accordance with the referees' wishes, we have now changed this sentence to “Distances and angles for these hydrogen bonds for 2d/2a are [C···O 3.17985(4)/3.199(2)Å, H···O 2.40149(4)/2.3041(13)Å, ∠C-H···O 138.9652(8)/161.19(11)°] for first pair of hydrogen bonds, and [C···O 2.99820(6)/3.078(2)Å, H···O 2.35724(4)/2.4076(15)Å, ∠C-H···O 124.3966(10)/128.91(11)°] for the second pair.”

Reviewer #1: line 143-144 - please check and correct the description of pi-pi interaction in structure 2a.

A: The sentence has been corrected to read:The crystal structure of 2a (Fig. 3) is strengthens by π-π interactions between benzene rings of the tosyl fragments of ligands with centroid-centroid distance 3.927 and 1.904 Å shift distance.

Reviewer #1: Figure 3 does not show the true packing along the crystallographic a axis (only 4 complex-molecules are used); the ellipsoids has no special meaning here, since they are defined in Fig. 2.

A: As suggested by the reviewer, we have revised Figure 3.

Reviewer #1: the citation [44] for SHELXL is wrong.

A: The new citation[45] Sheldrick, G.M. Crystal structure refinement with SHELXL. Acta Crystallographica Section C: Structural Chemistry, 2015, 71(1), 3-8. DOI:10.1107/S2053229614024218.

Reviewer 2 Report

Some of the experimental work has been done and described in a reliable way.

However, the work resembles a protocol from an experiment rather than a scientific article, as Authors’ own research results have unfortunately not been discussed with the literature data.  Please complete the manuscript with a thorough discussion of the results in each subsection of the work.

Pease also rewrite your Applications as the current version is a duplicate of the summary. In the Introduction, the authors wrote about the aim of the work: "The main goal of this work was to investigate the influence of the halogen substituent in the amine and aldehyde fragments in zinc complexes on their photoluminescent properties and biological activity", but unfortunately reading the Conclusions it is not known whether the goal has been achieved and what is the result of the research.

I also ask The Authors  to scan the protocols of all elemental analyzes and include them in the Supplementary Materials

Author Response

We thank the Referee for their interest in our work and for helpful comments and constructive suggestions that will greatly improve the quality of this manuscript. As indicated below, we have checked all the general and specific comments provided by the Referee and have made necessary changes according to their indications (the reviewer’s comments are in italics).

Reviewer #2: Some of the experimental work has been done and described in a reliable way. However, the work resembles a protocol from an experiment rather than a scientific article, as Authors’ own research results have unfortunately not been discussed with the literature data.  Please complete the manuscript with a thorough discussion of the results in each subsection of the work.

A: The discussion has been added to the manuscript.

Reviewer #2: Pease also rewrite your Applications as the current version is a duplicate of the summary. In the Introduction, the authors wrote about the aim of the work: "The main goal of this work was to investigate the influence of the halogen substituent in the amine and aldehyde fragments in zinc complexes on their photoluminescent properties and biological activity", but unfortunately reading the Conclusions it is not known whether the goal has been achieved and what is the result of the research.

A: In new Conclusion summarizes the results of our work, which indicates the effect of substitution of chlorine atoms in the aniline and aldehyde fragments of ligands of zinc complexes on their photoluminescent properties and biological activity.

Reviewer #2: I also ask The Authors to scan the protocols of all elemental analyzes and include them in the Supplementary Materials

A: The protocols of all elemental analyzes of azomethines and complexes include in the Supplementary Materials.

Reviewer 3 Report

The manuscript conducted studies on structure, photoluminescent properties and biological activities about zinc(II) complexes of azomethines ligands. Based on their previous work, a series of zinc-chelating compounds of azomethine ligands with or without chlorine was synthesized, and the effect of the halogen substituent on their photoluminescent properties was discussed. Moreover, most compounds had good protistocidal and antibacterial activities. Despite of these interestingly results, the manuscript needs minor revisions. The comments are provided below:

1.       In the part of “Abstract”, change “The structure of azomethines and their complexes was” into “The structures of azomethines and their complexes were”; change “It was found that” into “It is found that”.

2.       In the part of “Abstract”, the contents need be refined and improved.

3.       In the part of “Introduction”, compared with 2-hydroxybenzaldehydes derivatives reported by previous work, what are the advantages or differences of 2-(N-tosylamino)benzaldehyde and its derivatives as ligand?

4.       In the part of single-crystal X-ray diffraction, please marked shift distance of the centroid-centroid distance as 3.927 and 1.904 Å in Figure 3 with dashed lines.

5.       I observed that the author sometimes uses the term " UV-vis " and other instances " UV-Vis." Correct it throughout the text, please.

6.       In the Figure 4, change “Normalized emission spectra” into “Normalized absorption and emission spectra”.

7.       In the Figure 4, please provide the photographs of solid form of complexes before and after irradiation at 365 nm.

8.       In the biological activity, authors are advised to discuss the influence of chlorine substituent in more detail. Moreover, please supplement the comparative data of 1a-1e and 2a-2e in the part of discussion.

Author Response

We thank the Referees for their interest in our work and for helpful comments and constructive suggestions that will greatly improve the quality of this manuscript. As indicated below, we have checked all the general and specific comments provided by the Referees and have made necessary changes according to their indications (the reviewer’s comments are in italics).

Reviewer #3: The manuscript conducted studies on structure, photoluminescent properties and biological activities about zinc(II) complexes of azomethines ligands. Based on their previous work, a series of zinc-chelating compounds of azomethine ligands with or without chlorine was synthesized, and the effect of the halogen substituent on their photoluminescent properties was discussed. Moreover, most compounds had good protistocidal and antibacterial activities. Despite of these interestingly results, the manuscript needs minor revisions.

Many thanks to the reviewer for this positive feedback, which we appreciate.

Reviewer #3: In the part of “Abstract”, change “The structure of azomethines and their complexes was” into “The structures of azomethines and their complexes were”; change “It was found that” into “It is found that”.

A: We apologize for these errors,and we have corrected thetext as suggested.

Reviewer #3: In the part of “Abstract”, the contents need be refined and improved.

A: The respective changes have been made.

Reviewer #3: In the part of “Introduction”, compared with 2-hydroxybenzaldehydes derivatives reported by previous work, what are the advantages or differences of 2-(N-tosylamino)benzaldehyde and its derivatives as ligand?

A: Comparison of characteristics of 2-hydroxybenzaldehyde and 2-(N-tosylamino)benzaldehyde derivatives and their complexes added in the part of “Introduction”. In this work, we showed the differences in the structure, photoluminescence, and biological activity due to chlorine substituents, which can be used for fine-tuning of the properties of azomethine ligands.

Reviewer #3: In the part of single-crystal X-ray diffraction, please marked shift distance of the centroid-centroid distance as 3.927 and 1.904 Å in Figure 3 with dashed lines.

A: The proposed change has been made in Figure 3.

Reviewer #3: I observed that the author sometimes uses the term "UV-vis " and other instances " UV-Vis." Correct it throughout the text, please.

A: The correction has been made.

Reviewer #3: In the Figure 4, change “Normalized emission spectra” into “Normalized absorption and emission spectra”.

A: The proposed changes have been made to the caption to Figure 4.

Reviewer #3: In the Figure 4, please provide the photographs of solid form of complexes before and after irradiation at 365 nm.

A: The photographs of solid form of complexes before and after irradiation at 365 nm were added in the Figure 5.

Reviewer #3: In the biological activity, authors are advised to discuss the influence of chlorine substituent in more detail. Moreover, please supplement the comparative data of 1a-1e and 2a-2e in the part of discussion.

A: The influence of the chlorine substituent on the biological activity of 1a-1e and 2a-2e was considered in detail in the discussion part. This discussion has been added to the manuscript.

Round 2

Reviewer 2 Report

The authors ignored almost all of my comments.

The authors did not include protocols from elemental analyzes in Supplementary Materials.

They did not include a discussion of their own results with the available scientific literature; they only described more thoroughly their own biological research results (this is at the request of another reviewer).

The Conclusions are still missing " (…) the influence of the halogen substituent in the amine and aldehyde fragments in zinc complexes  (…) on biological activity".

Author Response

IJMS-1941620, «Synthesis, structure, spectral-luminescent properties, and biological activity of chlorine-substituted N-[2-(phenyliminomethyl)phenyl]-4-methylbenzenesulfamide and their zinc(II) complexes»

Anatolii S. Burlov, Valery G. Vlasenko, Maxim S. Milutka, Yurii V. Koshchienko, Nadezhda I. Makarova, Vladimir A. Lazarenko, Alexander L. Trigub, Alexandra A. Kolodina, Alexander A. Zubenko, Anatoly V. Metelitsa, Dmitrii A. Garnovskii, Alexey N. Gusev and Wolfgang Linert

ANSWERS TO REVIEWERS

We thank the reviewer for these helpful comments but we do not agree with their general statement by the following reason.

Reviewer #2: The authors ignored almost all of my comments.

A: The authors in the revised manuscript tried to take into account all the comments of the reviewer. The discussion of the effect of substitution of chlorine atoms in the aniline and aldehyde fragments of ligands of zinc complexes on their photoluminescent properties and biological activity has been added to the manuscript.

Reviewer #2: The authors did not include protocols from elemental analyzes in Supplementary Materials.

A: The protocols of all elemental analyzes of azomethines and complexes are included in the Supplementary Materials. Please note that the initial protocols of elemental analysis were obtained by the authors from a specialized laboratory and represent standard tables with data in Russian. Perhaps the reviewer did not pay attention to these protocols in the Supplementary Materials.

Reviewer #2: They did not include a discussion of their own results with the available scientific literature; they only described more thoroughly their own biological research results (this is at the request of another reviewer). The Conclusions are still missing " (…) the influence of the halogen substituent in the amine and aldehyde fragments in zinc complexes  (…) on biological activity".

A: The authors rewrote the section on biological activity in accordance with the reviewer's comments, where they considered in detail the effect of chlorine substituents on biological activity in a number of studied compounds, and in comparison with previous data for complexes of salicylic aldehydes. In addition, this material about biological activity of complexes is additional, since the main goal of the work is to study the structure and photoluminescent properties of these compounds (which is consistent with the subject of the journal issue).